# Nutritional Prehabilitation in Patients Undergoing Abdominal Surgery—A Narrative Review

**DOI:** 10.3390/nu16142235

**Published:** 2024-07-11

**Authors:** Maria Wobith, Aileen Hill, Martin Fischer, Arved Weimann

**Affiliations:** 1Department for General, Visceral, and Oncological Surgery, Klinikum St. Georg, 04129 Leipzig, Germany; maria.wobith@gmx.de (M.W.); martin.fischer@sanktgeorg.de (M.F.); 2Department for Operative Intensive Care and Intermediate Care, University Hospital RWTH Aachen, 52074 Aachen, Germany; ahill@ukaachen.de; 3Department for Anaesthesiology, University Hospital RWTH Aachen, 52074 Aachen, Germany

**Keywords:** nutritional prehabilitation, malnutrition, gastrointestinal surgery, medical nutrition therapy, perioperative, literature review

## Abstract

Malnutrition plays a crucial role as a risk factor in patients undergoing major abdominal surgery. To mitigate the risk of complications, nutritional prehabilitation has been recommended for malnourished patients and those at severe metabolic risk. Various approaches have been devised, ranging from traditional short-term conditioning lasting 7–14 days to longer periods integrated into a comprehensive multimodal prehabilitation program. However, a significant challenge is the considerable heterogeneity of nutritional interventions, leading to a lack of clear, synthesizable evidence for specific dietary recommendations. This narrative review aims to outline the concept of nutritional prehabilitation, offers practical recommendations for clinical implementation, and also highlights the barriers and facilitators involved.

## 1. Introduction

Prehabilitation aligns with the traditional approach of conditioning patients before surgery to reduce postoperative complications and improve outcomes. Malnutrition is recognized as a modifiable risk factor prior to surgery [1]. Additionally, reduced muscle mass, sarcopenia, and sarcopenic obesity are considered to be modifiable risk factors as well [2]. The guidelines from the European Society for Clinical Nutrition and Metabolism (ESPEN) regarding Clinical Nutrition in Surgery recommend delaying surgery for severely malnourished patients for 7–14 days to enhance their nutritional status [3]. There is evidence suggesting that extended preoperative conditioning over 2–6 weeks in high-risk patients not only enhances physical function and nutritional status but also positively impacts postoperative recovery [4].

The concept of prehabilitation seeks to optimize a patient’s functional status before surgery through a multidisciplinary, personalized approach [4,5]. The primary goals include reducing surgery-related stress and temporary functional decline while ensuring adherence to the individually selected therapeutic measures for each patient, such as suggested in the current Enhanced Recovery After Surgery (ERAS) guidelines [5,6]. Additionally, in cancer patients, the carcinoma significantly affects immunomodulating processes, negatively impacting nutritional status and body composition. Therefore, most studies on prehabilitation programs focus on cancer patients undergoing surgery to mitigate inflammatory processes.

While patients may benefit from various interventions in the preoperative phase, the typical prehabilitation approach involves endurance and resistance training, nutritional therapy, and psychological coaching. A recent international Delphi survey identified optimizing nutritional regimens as a key research priority in prehabilitation [7].

However, barriers and facilitators for preoperative nutrition therapy within the prehabilitation framework have not been thoroughly explored, which are the focus of this narrative review.

Methodology: For this narrative review, a non-systematic literature search was conducted on PubMed in September 2023 using the terms “Prehabilitation” and “Nutrition”, as well as “ONS” (Oral Nutritional Supplements) and “Compliance”. We primarily included randomized controlled trials (RCTs), meta-analyses, and systematic reviews. Emphasizing psychological aspects, several recommendations for clinical practice were formulated. Furthermore, guidelines recommendations were consulted to provide a comprehensive overview.

## 2. Nutritional Prehabilitation—Current Evidence

Given that nutritional status significantly impacts the short- and long-term outcomes of surgical patients, it is reasonable, and may be even mandatory, to identify malnutrition preoperatively [3]. Early identification of patients requiring nutritional therapy is crucial, and nutritional prehabilitation should be considered. It is important to note that malnutrition in overweight or obese patients is frequently overseen, particularly in cases of gastrointestinal (GI) cancer, where sarcopenic obesity is a concern [8].

Sarcopenia, reduced muscle mass, and malnutrition have been shown to negatively impact the postoperative course regarding the occurrence of complications and length of hospital stay [3,9,10]. Additionally, nutrition therapy is a cornerstone of ERAS protocols, commencing before surgery to optimize metabolic health, improve GI function and increase compliance with the ERAS program [6].

Determining the optimal timing to initiate nutritional prehabilitation is important in all patients undergoing abdominal surgery but is a key aspect for GI cancer patients as part of the comprehensive multimodal treatment strategy. Neoadjuvant chemo- or radiochemotherapy provides a window of opportunity for supportive interventions before surgery. The early initiation of a carefully planned and individually adapted medical nutrition therapy (MNT) may prevent significant preoperative weight loss and perioperative functional decline, as demonstrated in studies involving patients undergoing esophageal resection [11].

The prevalence of six modifiable risk factors for perioperative outcomes was assessed recently, and the need for specific interventions within a prehabilitation program for patients undergoing hepatopancreaticobiliary cancer surgery was determined [12]. Malnutrition, as diagnosed by patient-generated subjective global assessment (PG-SGA), was present in 42% of patients and prompted nutritional intervention, highlighting the importance of this topic.

## 3. Unimodal Nutritional Prehabilitation

There is a lack of definitive evidence regarding the optimal duration and type of nutritional prehabilitation, considering the different patient groups and possible interventional strategies. 

In a meta-analysis of 33 studies involving elderly individuals undergoing major abdominal surgery (n = 3962), nutritional prehabilitation demonstrated superiority over standard care [13]. Most of the included RCTs were conducted prior to the emergence of prehabilitation programs, involving unimodal preoperative nutrition interventions lasting 7 to 10 days. This meta-analysis revealed a decrease in postoperative complication rates through targeted preoperative nutritional therapy (risk difference −0.18, 95% CI −0.26 to −0.10; *p* < 0.001, I^2^ = 0%) [13]. All the examined nutrition-only prehabilitation programs included ONSs, with intake ranging from ad libitum to 400 mL three times a day and prescribed durations varying from one week to four weeks. These programs demonstrated significantly fewer overall complication rates compared to standard care. Another meta-analysis by Gillis et al., with patients undergoing colorectal surgery, revealed that nutritional prehabilitation with and without exercise significantly decreased the length of hospital stay (weighted mean difference in the length of hospital stay = −2.2 days; 95% CI −3.5 to −0.9) [14]. However, additional exercise may accelerate the return to presurgical functional capacity [14].

However, data inconsistencies make recommendations contentious: Another meta-analysis focusing on patients with colorectal cancer receiving preoperative ONSs did not reveal differences in postoperative outcomes [15]. High heterogeneity among the included studies was a notable issue, with malnutrition prevalence ranging from 8% to 68% and patient compliance ranging from 72% to 100%.

Hence, it comes as no surprise that McIsaac et al. found, in an umbrella review encompassing 55 meta-analyses including patients undergoing any surgery, relatively low to very low evidence supporting the effectiveness of nutritional prehabilitation in reducing complications, mortality, and the length of hospital stay [16].

## 4. Nutrition as Part of Multimodal Prehabilitation

Currently, MNT in prehabilitation programs is rarely used unimodally. From a physiological standpoint, combining nutrition and exercise is recommended due to the combined anabolic stimuli. Most studies have analyzed the combination of nutrition therapy and exercise, observing improvements in physical function [6-min walking distance: in the preoperative period: 30.0 (SD 46.7) m vs. −5.8 (SD 40.1) m, *p* < 0.001; at four weeks: −11.2 (SD 72) m vs. −72.5 (SD 129) m, *p* < 0.01; at eight weeks: 17.0 (SD 84.0) m vs. −8.8 (SD 74.0) m, *p* = 0.047 [6]. As demonstrated in recent systematic reviews and meta-analyses, multimodal prehabilitation will primarily improve functional capacity and physical performance before surgery [17,18]. A significant decrease in the complication rate may be awaited with special regard to frail and high-risk patients [19].

Comparing nutritional prehabilitation alone with combined exercise and nutrition, it has been demonstrated that each modality alone reduces the length of hospital stay [14]. However, multimodal prehabilitation further enhances functional capacity, such as the 6-min walking distance [14]. Nevertheless, an umbrella review revealed that combined exercise and nutritional prehabilitation is shown to be effective, but the effect remains low to very low [16].

A recently published trimodal prehabilitation program designed for patients undergoing colorectal resection lasted for over four weeks [20]. This program included nutrition therapy with dietary guidance aimed at achieving a protein intake of 1.5 g/kg body weight, along with a whey protein supplement (30 g) consumed one hour after exercise and one hour before bedtime, as well as a multivitamin supplement. The results of the study demonstrated a significantly lower incidence of severe complications (CCI score >20) in the prehabilitation group compared to standard care (21 of 123 [17.1%] vs. 38 of 128 [29.7%]; OR 0.47 [95% CI 0.26 to 0.87]; *p* = 0.02).

Nevertheless, evidence for giving specific recommendations regarding MNT in multimodal prehabilitation programs is sparse [13,14,21]. Consequently, the specific types of nutritional interventions may be subject to debate. Although most randomized studies have been conducted for durations of 7–14 days, there are no data available for periods ranging between four and six weeks. In comparison with standard ONSs, preoperative immunonutrition and synbiotics have been shown to be beneficial, significantly reducing complications when administered for 5–7 days prior to surgery [22,23].

In summary, the heterogeneity of study protocols (including primary endpoints and study populations) and variations in prehabilitation programs themselves (such as types of interventions, duration, and intensity) pose challenges to comparability in meta-analyses. 

## 5. Assessment of Nutritional Status and Indication for Prehabilitation

Recognizing the critical importance of patients’ nutritional status, standardized nutritional care has been implemented for nutritional prehabilitation in oncology patients [24]. This approach comprises four key steps: Nutrition assessment;Nutrition diagnosis;Nutrition therapy;Nutrition monitoring and evaluation.

Screening for nutritional status should be conducted in all hospitalized patients during the initial contact using a validated tool. The Nutritional Risk Score-2002 (NRS-2002) has been well validated for surgical patients [24] and is rapidly performed in clinical routine. Furthermore, the Subjective Global Assessment (SGA) or the Patient-generated SGA (PG-SGA) is used for malnutrition screening and assessment in clinical studies. A positive screening result should prompt a comprehensive assessment. Even in cases of negative screening results, subsequent screenings should be performed during follow-up visits.

In several studies, particularly those employing a multimodal approach, regular assessment of nutritional status before prehabilitation has not been consistently implemented, despite all patients receiving standardized therapy such as ONSs [13]. While this leads to greater generalizability of the results due to broad inclusion of patients, selecting and treating only patients in need of medical nutrition therapy would be more efficient.

In 2017, ESPEN defined severe metabolic risk if one of the following criteria is present [2]: Weight loss > 10–15% within 6 months;Body Mass Index (BMI) < 18.5 kg/m^2^;Subjective Global Assessment (SGA) grade C, NRS-2002 > 5;Serum albumin < 30 g/L (ruling out liver or kidney failure).

These parameters encompass both, nutritional status and disease-associated malnutrition. Ideally, for oncologic patients, access to computerized tomography (CT) scans is advantageous, as they can be utilized for body composition analysis. In an axial CT scan at the lumbar level 3 during the portal vein phase, muscle mass and muscle quality can be measured using various software programs based on tissue-specific attenuation values for skeletal muscle (−29 to 150 Hounsfield Units) [25]. The skeletal muscle area in this slice includes seven muscle groups: the psoas, erector spinae, quadratus lumborum, transversus abdominis, external obliques, internal obliques, and rectus abdominis. The Skeletal Muscle Index (SMI), which is used for diagnosing sarcopenia, can be easily calculated by dividing the skeletal muscle area by height squared. The CT-derived SMI can then be integrated into the GLIM (Global Leadership Initiative for Malnutrition) criteria for diagnosing malnutrition as well [26]. Studies have shown that CT-derived reduced muscle mass, as per GLIM criteria, correlates with surgical outcomes in patients undergoing major abdominal surgery [27].

Currently, it is recommended to postpone surgery for severely malnourished patients for at least 7–14 days to allow sufficient time for prehabilitation measures to take effect. However, decisions regarding postponement require an interdisciplinary approach and effective communication with the patient. For certain patient groups, particularly those with malnutrition, sarcopenia, or frailty, international guidelines suggest postponing surgery for 4–6 weeks to ensure adequate time for prehabilitation [8]. Studies have demonstrated that implementing these measures in patients undergoing elective surgery reduces surgical stress and catabolism, promoting the attainment of an anabolic state. This, in turn, facilitates better and faster recovery from major surgical procedures. Furthermore, research by the working group of Gillis and Carli suggests that prehabilitation may be more effective than rehabilitation alone for patients undergoing abdominal surgery [9]. Ideally, prehabilitation and ERAS programs should complement each other, incorporating interdisciplinary approaches such as multimodal analgesia, optimized nutrition, exercise, and effective management of postoperative nausea and vomiting.

## 6. Duration of Nutritional Prehabilitation

During the preoperative period, many patients fail to meet their energy and protein requirements due to anorexia and the catabolic state induced by inflammatory diseases. This is particularly prevalent in patients with upper GI cancer and those undergoing neoadjuvant chemotherapy, who often experience loss of appetite, nausea, or vomiting, leading to reduced oral food intake [28]. 

Yamamoto et al. demonstrated that implementing a prehabilitation program with exercise and nutritional support for elderly sarcopenic patients with gastric cancer significantly increased calorie and protein intake [29.4 ± 6.9 kcal/kg ideal body weight (IBW) vs. 27.3 ± 5.6 kcal/kg IBW, *p* = 0.049, and 1.3 ± 0.4 g/kg IBW vs. 1.1 ± 0.3 g/kg IBW, *p* = 0.0019, respectively] [29]. Although the increase in SMI observed in this study was not statistically significant, at least four patients transitioned from being sarcopenic to nonsarcopenic after completing the program. Given that the median duration of prehabilitation in this study was just 16 days, it is possible that longer treatment periods could further improve the observed effects.

Another study focusing on gastric cancer patients investigated the impact of preoperative nutrition therapy in malnourished patients [30]. It was found that therapy lasting at least 10–13 days resulted in a significant decrease in the number of surgical site infections (17.0 vs. 45.4%; *p* = 0.0006). Regarding the timing of protein intake, it is recommended to consume a high-protein diet after exercise to optimize muscle protein synthesis based on physiological principles [31].

## 7. Nutritional Interventions Used for Prehabilitation

### 7.1. ONSs

According to ESPEN, “ONSs are developed to provide energy and nutrient-dense solutions” [32] and “as an appropriate measure to increase energy and protein intake in case normal oral food intake is not meeting requirements”. ONSs are classified as Food for Special Medical Purposes under European Union regulation 609/2013.

In terms of composition, ONSs are typically rich in energy and protein, although their specific contents can vary significantly depending on the product. Many ONS products boast a balanced or complete nutrient profile, enabling them to serve as the sole source of nutrition for an extended period if consumed in adequate quantities, typically around 4–5 units per day [32].

Standard ONSs range between 150 and 300 mL per unit and are intended to be consumed between 2 and 5 times a day between meals, or as late-night snacks to prevent the patients from feeling overly full during regular mealtimes. Patients are advised to consume ONSs slowly (“sip-feeding”) to minimize GI discomfort. 

Some ONS formulations may lack certain nutrients and may be enriched with vitamins, trace elements, or specialized fatty acids to meet specific nutritional needs, as summarized in Table 1. However, nutritionally incomplete ONSs should not be relied upon as the sole source of nutrients [32]. The choice of ONSs should be adapted to the individual patient’s preferences and abilities. A large range of styles, textures and flavors are available, as displayed in Table 2.

Additionally, in case of deficiency, the supplementation of vitamins and micronutrients can be considered. The most important to mention are vitamin D, folate, and vitamin B12. Some studies provided multivitamin supplements during prehabilitation [20].

### 7.2. Immunonutrition

Utilizing the terms “immo- or immunonutrition”, ONSs designed to enhance the immune system have been developed to modulate the perioperative stress response in patients undergoing major surgery. A combination of arginine, omega-3 fatty acids, and ribonucleotides has been extensively investigated in numerous RCTs and meta-analyses, evaluated within an umbrella review of meta-analyses [17], and remains a subject of ongoing debate. In comparison with standard ONSs, preoperative immunonutrition and synbiotics have been shown to be beneficial, significantly reducing complications when administered for 5–7 days prior to surgery [22,23].

The timing of immunonutrition remains a persistent question. A meta-analysis comprising data from 16 randomized studies involving 1387 surgical patients with GI tumors (715 received immunonutrition, 672 served as controls) focused on the exclusive preoperative administration for 5 to 7 days [33]. Significantly lower incidences of infectious complications compared to a normal diet and an isonitrogenous standard ONS (OR 0.52, 95% CI 0.38–0.71, *p* < 0.0001) were revealed, with a low heterogeneity of data (I^2^ = 16%). Additionally, there was a notable reduction in the length of hospital stay compared to a normal diet and a tendency compared to the standard ONS (−1.57 days, 95% CI −2.48–−0.66, *p* = 0.0007, I^2^ = 34%). No significant differences were observed between groups in terms of non-infectious complications and mortality. Therefore, it can be inferred that the preoperative administration of immunonutrition for five to seven days may be effective and superior to the standard ONS in this patient group.

A more recent meta-analysis, encompassing 37 RCTs with 3793 patients, found a reduction in postoperative infectious complications with immunonutrition (OR 0.58, 95% CI 0.47–0.72). This association was significant only in subgroup analyses with preoperative and perioperative administration and in trials including upper GI cancers, colorectal cancer, and “mixed GI” cancer populations. Significance was independent of nutritional status [34].

Focusing on patients undergoing esophageal resection, another meta-analysis of 15 RCTs showed that immunonutrition was superior to standard nutrition in terms of reducing infectious complications and length of hospital stay [9].

Immunonutrition within an ERAS program was examined in the SONVI study, which randomized 264 patients undergoing colorectal resection [35]. Comparing immunonutrition to a control group receiving hypercaloric hypernitrogenous supplements for seven days before and until five days after surgery, no differences in length of stay were observed. However, patients receiving immunonutrition experienced a decrease in the number of complications, primarily due to a significant reduction in infectious complications (23.8% vs. 10.7%, *p* = 0.0007).

The current ESPEN guideline recommends the intake of ONSs before major abdominal surgery for 5–7 days, with immunomodulating supplements being preferred [3].

## 8. Barriers and Facilitators for Nutritional Prehabilitation

Nutrition therapy within prehabilitation programs faces inherent challenges regarding heterogeneity and standards [36]. Many studies lack adequate assessment of nutritional status, particularly within multimodal programs.

A recent scoping review of nutrition in prehabilitation oncology research found inconsistencies in nutrition assessment, interventions often falling short of reference standards, and two-thirds of the reviewed studies failing to monitor the nutrition intervention or evaluate nutrition outcomes [36].

In an RCT involving frail geriatric patients undergoing cancer surgery, qualitative interviews investigated facilitators and barriers for exercise prehabilitation [37]. Facilitators included program manageability and suitability for older adults with frailty, adequate resources for engagement, support from others, a sense of control, intrinsic value, progress recognition, improved health outcomes, and enjoyment facilitated by prior experience. Barriers encompassed pre-existing conditions, fatigue, baseline fitness, weather, and feelings of guilt and frustration when unable to exercise [37]. There was consensus regarding the need for individualization and variety.

Considering some overlap with nutritional prehabilitation, specific barriers and facilitators will be addressed in Figure 1. 

### 8.1. Specific Barriers

#### 8.1.1. Delay of Surgery

Depending on the individual patient’s trajectory, it may be prudent to extend the period of prehabilitation and defer major treatment such as surgery, as depicted in Figure 2. Such decisions naturally necessitate an interdisciplinary approach and effective communication with the patient. In certain patient groups, particularly those with malnutrition, sarcopenia, or frailty, current international guidelines recommend postponing surgery for four to six weeks to ensure ample time for prehabilitation [8].

The existing challenges related to heterogeneity in the duration and specific interventions within nutritional prehabilitation continue to pose a barrier. The current literature does not provide specific recommendations due to this variability, particularly in the context of multimodal prehabilitation programs lasting longer than 1–2 weeks. The decision to postpone surgery for several weeks in cancer patients may be subject to debate. However, data from colorectal cancer patients have demonstrated significantly improved 5-year disease-free survival in Union for International Cancer Control (UICC) stage III after prehabilitation [38].

In cases involving neoadjuvant treatment, there is a time window of several weeks before surgery, during which prehabilitation can be implemented without delaying the surgical procedure. Prehabilitation can also commence at the onset of therapy, as it has been shown to mitigate losses in functional capacity following neoadjuvant treatment [39,40]. Recently, data from the Swedish national register indicated that for patients with colon cancer, a period of up to 56 days from diagnosis to the initiation of curative treatment is not associated with worse overall survival [41].

#### 8.1.2. Unclear Responsibilities—Who Is in Charge?

The responsibility for nutrition therapy, including early involvement of a nutrition support team, is often poorly organized in the preoperative period. While the diagnosis of the underlying disease, such as cancer, may be initiated by the gastroenterologist, the surgeon and anesthesiologists are responsible for the patient’s care during the perioperative period, ensuring the patient’s safety throughout the surgical procedure. Therefore, all medical staff involved in treating the patient should be aware of the patient’s nutritional status and how the individual medical nutrition therapy is organized. This responsibility can be carried out by the surgeons themselves or organized through an outpatient clinic, prehabilitation clinic, or in cooperation with the general practitioner. Viewing prehabilitation as a part of multimodal cancer therapy and including decisions within the tumor board is one approach to caring for each patient individually. However, this procedure depends strictly on the healthcare system and the organization within the individual hospital.

#### 8.1.3. Reimbursement

There is an ongoing debate surrounding the cost-effectiveness of using ONSs. Depending on the healthcare system, nutrition therapy involving dietary counseling and ONSs may not be routinely covered by health insurance. A meta-analysis of nine studies including 11 cost-analyses revealed cost savings associated with reductions in infectious complications and the length of hospital stay [42]. In Germany, standard ONSs may only be prescribed when disease-related malnutrition has been clearly diagnosed and all efforts to improve oral intake through conventional food, including fortification, have been exhausted. It is evident that a cost-effective approach should involve both preventing the deterioration of nutritional status and treating malnutrition.

#### 8.1.4. GI Symptoms, Tolerance, and Compliance

Compliance with the intake of ONSs may be hindered primarily due to issues such as taste, bloating, or diarrhea. In patients undergoing colorectal cancer surgery, compliance with preoperative ONS intake ranged from 72 to 100% [15]. Limitations primarily stem from a lack of patient education, for example, drinking a full bottle versus sip-feeding and thus inducing adverse effects. For patients with gastric cancer, Wan et al. observed an average compliance range of 31% over a 12-week period [43]. In interviews, the authors identified factors such as social support, adverse reactions, patient attitudes, and motivation as relevant to compliance with oral supplementation. Additionally, the primary caregivers and income were identified as independent factors related to compliance in patients with gastric cancer [43].

In a prehabilitation study involving patients undergoing pelvic exenteration, 12 out of 20 participants were non-compliant with the targeted intake [44]. Complaints associated with the intake of ONSs included flavor, volume, texture, impact on dietary intake, and motivation. Well-nourished patients exhibited higher compliance, highlighting the need for patient education and individual tailoring of the prescription to each patient’s needs.

It is also important to educate patients and caregivers that conditions such as diarrhea, loss of appetite, vomiting, nausea, or the symptoms of bowel obstructions can be caused by the disease itself, or by additional neoadjuvant treatment, and are not necessarily caused by the nutritional intervention. Several nutritional assessments inquire about these symptoms, and they must be treated symptomatically to achieve the optimal nutrient intake and successful prehabilitation. Additionally, careful dietary counseling aids in overcoming symptoms and finding individualized approaches to support oral intake and nutrition therapy [24].

A recent review of the literature sought to identify factors influencing patients’ adherence to ONS intake [45]. They discovered that sensory attributes, particularly palatability and consistency significantly influenced intake. An area that remains relatively understudied is the role of aroma, especially among older adults, which warrants further investigation to optimize intake.

Compliance with intake is often limited when it comes to consuming up to three times a day, 400 mL each time. To increase caloric intake, ONSs with higher energy density (1.5 or 2 kcal/mL) may be considered, but special emphasis must be placed on slow consumption despite low volume to avoid side effects [46]. Close monitoring of effects and adverse effects, along with repeated dietary advice and continuous patient motivation, may be necessary to maintain good compliance. The American Society of Parenteral and Enteral Nutrition (ASPEN) recommends that patients and dietitians discuss the amount and frequency of ONS intake, different flavors and types to ensure personal taste and preferences are considered, as well as any relevant food allergies or intolerances [47].

### 8.2. Specific Facilitators

#### 8.2.1. Adherence

The role of the nutrition support team and dietary counseling is crucial. ASPEN recommends that patients stay in contact with a dietitian regularly “so they can monitor and review your progress” [47]. This regular contact is essential for adherence. Thoughtful dietary counseling should be considered for patients undergoing major abdominal surgery. Traditionally, patients receive counseling towards the end of their hospital stay, but it is important to start this process early on, at the beginning of their care journey to identify and treat problems early on while patients are still in the hospital [48,49]. One approach to increase ONS intake in the hospital setting is to administer them during medication rounds. In a recent RCT (MEDPass), patients received 50 mL of ONSs four times per day during medication rounds or through conventional supply [50]. While overall compliance with ONS intake was high, there was no significant improvement observed when ONSs were administered during medication rounds. This highlights the need for nutrition therapy to be tailored to each patient’s individual needs and challenges.

Recent research has shown that high-protein, low-volume ONSs consumed twice daily allows most cancer patients to meet minimal ESPEN protein recommendations [51]. Additionally, so far unpublished results from our own study involving high-risk patients undergoing multimodal prehabilitation for six weeks before major abdominal cancer surgery after neoadjuvant treatment have shown an adherence of over 80% to every component of the program, including twice-daily intake of ONSs (20 g protein, 300 kcal in 200 mL).

#### 8.2.2. Psychological Factors

Many of the psychological factors that are relevant with respect to participation in nutritional prehabilitation are related to adherence, i.e., a patient’s level of readiness and commitment to implement the intended nutritional changes. Internal motivation, social support, and self-efficacy, for example, facilitate participation and can also be promoted throughout the prehabilitation program (Table 3). Other factors such as conscientiousness or cognitive flexibility are less subject to change but help to establish a personalized approach for improving adherence in conversations. Depressed mood is a critical psychological barrier for participation and usually requires professional psychological support. Ideally, such a support is available on demand. Evidence of specific psychological interventions is lacking due to heterogeneity of studies [52]. Unimodal psychological prehabilitation has not yet been shown to be efficient, but in a multimodal approach, it is recommended to support other interventions like exercise and nutrition therapy. Common psychological facilitators and barriers as well as respective recommendations are summarized in Table 3. More detailed recommendations can be derived from a list of 93 behavior change techniques [53].

#### 8.2.3. Monitoring and Supervision

As discussed earlier, nutritional prehabilitation heavily relies on patient compliance and adherence, which necessitates individual monitoring of tolerance [24]. Therefore, it is essential to track weight changes, gastrointestinal symptoms, and oral food intake during prehabilitation to adjust nutrition therapy and address individual issues [48,49]. Nutrition therapy without monitoring at the outset may not be advisable, especially if the duration exceeds one week; regular monitoring and encouragement is necessary. This should involve at least a phone call to inquire about specific problems, current weight, tolerance, and GI symptoms [47]. Additionally, future smartphone applications could facilitate remote monitoring, potentially even allowing self-evaluation by the patient [54]. During phases of neoadjuvant treatment, it is crucial to reassess body composition to identify any deterioration in nutritional status and adjust or modify nutrition therapy accordingly. Alongside weight measurement, bioelectrical impedance analysis (BIA), which is easily conducted, may be more suitable for making improvements in nutritional status visible to the patient.

A limitation of this narrative review is that we did not conduct a systematic literature review or meta-analysis. Narrative reviews have inherent limitations in terms of objectivity, completeness of the literature search, and interpretation of findings.

## 9. Conclusions

Nutritional prehabilitation, alone or in a multimodal approach, has been demonstrated to be effective regarding patient function and outcome as well as for the cost–benefit ratio. While specific recommendations may vary due to study heterogeneity, strategies like preoperative intake of ONSs have shown efficacy especially in a multimodal setting. To overcome obstacles and promote nutritional prehabilitation, key steps for ensuring patient engagement include the following:Implementing a well-structured prehabilitation program that offers personalized dietary counseling and monitoring.Motivating the patient through cognitive behavioral psychology strategies.

## Figures and Tables

**Figure 1 nutrients-16-02235-f001:**
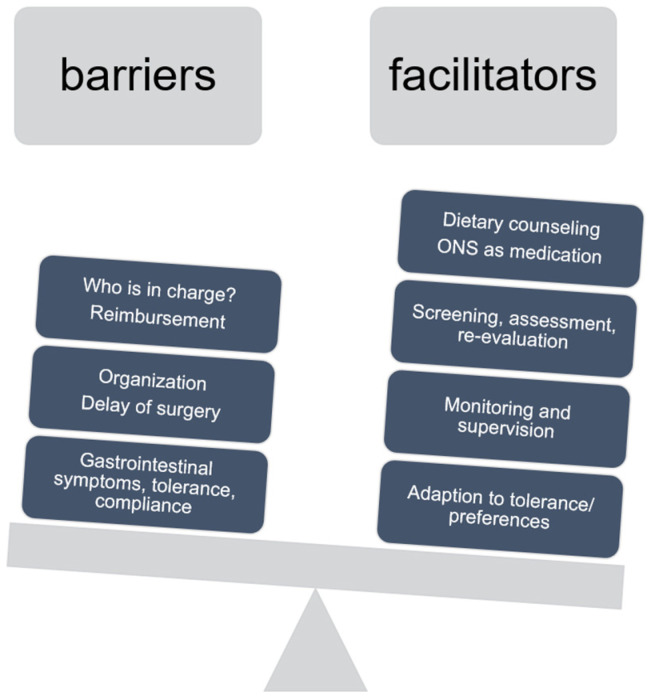
Overview on barriers and facilitators of nutritional prehabilitation.

**Figure 2 nutrients-16-02235-f002:**
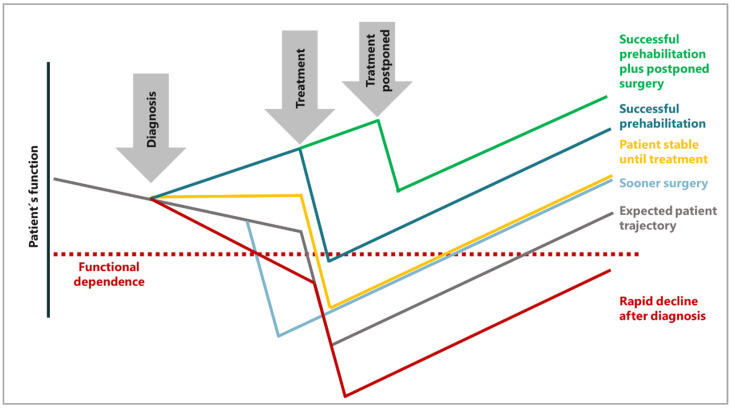
Concept of different patient trajectories, modified from Carli et al. [6].

**Table 1 nutrients-16-02235-t001:** Examples of different ONS compositions.

ONS Category	Component Altered	Examples	Suggested for Malnourished Patients with
Macronutrient composition	Volume to macronutrient ratio	High energy High protein	Heart or kidney failure, respiratory diseases
	Fatty acids	EHA, DPA Omega-3 fatty acids	Oncologic diseases Impaired wound healing
	Amino acids	Branched chain amino acids (BCAA)	Patients with liver failure
		Single amino acids (e.g., HMB, leucine, arginine, glutamine)	Patients with too little protein intake
		Whey protein	
	Carbohydrates	Reduced carbohydrates	Diabetes Chronic respiratory failure
	Fiber (prebiotics)		Stool frequency irregularities
Micronutrient composition	Vitamins		
	Electrolytes		Chronic renal failure
	Trace elements		Bone health diseases
Probiotic additions	Probiotics		
	Synbiotics		

**Table 2 nutrients-16-02235-t002:** Examples of different ONS styles accommodating patients preferences and abilities.

Styles and Textures	Milk shakes Juices Puddings and yogurts Soups Powder
Flavor	Neutral Sweet (chocolate, vanilla, nut, cappuccino, coffee), Fruity (mango, apricot, raspberry, red fruit, orange, apple, banana), Hearty (tomato, mushroom, asparagus, vegetables, chicken, curry, pumpkin, carrot)

**Table 3 nutrients-16-02235-t003:** Psychological facilitators and barriers of nutritional prehabilitation participation.

Facilitator/Barrier	Recommendation
Internal motivation	Introduce potential benefits of participation in a personalized way (e.g., link to individual needs and values) Educate on the rationale of each encouraged change throughout the entire prehabilitation period Adjust goals and demands to patients’ abilities and resources (e.g., time, space, and finances) Offer options and alternative choices Provide regular and individualized feedback
Self-efficacy	Work with persuasion (e.g., verbal encouragement based on past achievements) Introduce role models (e.g., via a support group) Set progressive goals
Depressed mood	Provide on-demand psychological support (e.g., cognitive behavioral counseling) Refer to treatment in case of clinical depression (e.g., psychotherapy, antidepressants)
Social support	Involve relatives early and throughout the entire prehabilitation process as facilitators (e.g., for cheering, moral support, food preparation, support on other daily duties) Provide face-to-face contact with a dietitian or other professional Offer direct and accessible communication (e.g., chat, phone)
Conscientiousness	Assess and discuss level of commitment Increase commitment by providing resources and aids (e.g., free samples and supplies)
Cognitive flexibility	Address and dispute dysfunctional beliefs regarding prehabilitation participation (e.g., “I’m too old to change my diet”, “it is going to be a hassle”) In case of inflexibility, emphasize planned changes as modes to maintain current habits (e.g., “You’ve always eaten fish on Fridays? Excellent please keep it that way, as it will help you to increase protein intake on that day. We will just add a little more of the fish.”)

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
