# Peer review of "Nutritional Prehabilitation in Patients Undergoing Abdominal Surgery—A Narrative Review"

_nutrients, 2024, doi:10.3390/nu16142235_

Round 1
Reviewer 1 Report
Comments and Suggestions for Authors
Dear authors,
I have carefully studied the manuscript entitled “Nutritional prehabilitation in patients undergoing abdominal surgery – a narrative review” by Maria Wobith et al.
The topic is very interesting but some adjustments need to be made:
1. line 130 - you have 2 point in the title
2. please add a diagram with the mechanism implicated in cashexia
3. please add the limitation section
Author Response
Dear Reviewer,
thank you for your remarks. We appreciate the suggested proposals, which we incorporated in the text.
Comments 1: line 130 - you have 2 point in the title
Response 1: Changed in every headline as there was an editorial error.
Comments 2: please add a diagram with the mechanism implicated in cashexia
Response 2: We appreciate the reviewer´s comment. Cachexia is not a focus of this manuscript and we have chosen not to add a figure or diagram for this very complex topic.
Comments 3: please add the limitation section
Response 3: We added a paragraph of the limitations in lines 463-465: A limitation of this narrative review is that we did not conduct a systematic literature review or meta-analysis. Narrative reviews have inherent limitations in terms of objectivity, completeness of literature search, and interpretation of findings.
Reviewer 2 Report
Comments and Suggestions for Authors
The authors have chosen to discuss in this review the rationale and the indication of Nutritional prehabilitation in patients undergoing abdominal surgery. The authors have provided important information regarding the evidence of nutritional prehabilitation alone or in combination with other prehabilitative strategies.
In addition to malnutrition, reduced muscle mass and sarcopenia are also considered modifiable risk factors that can be improved with nutritional therapy. I recommend adding these elements in the introduction section.
Line 55, page 2: add reduced muscle mass not only sarcopenia
In "Unimodal nutritional prehabilitation" section specify the type of surgery performed in the studies cited
The bibliography and consequently the text must be updated with recent systematic reviews such as: Mareschal et al. Cancer (2023), Jain et al. World J Surg (2023), Pavel Skorepa et al. Clinical Nutrition (2024)
In most published studies regarding surgical prehabilitation, the PG-SGA is used as a method of malnutrition assessment. Add it in the "Assessment of nutritional status and indication for prehabilitation" section. Body composition assessment and measurement methods must be clarified better in the "Assessment of nutritional status and indication for prehabilitation" section.
I recommend adding ONS containing whey proteins in table 1.
I recommend separating table 1 into two tables: a table relating to the composition of ONS and a table relating to Textures and flavours (the powdered ONS should also be added).
Vitamin supplements should also be considered in case of deficiencies (vitamin D, B12, folate...) as they are part of the nutritional intervention.
Most of the published studies included cancer patients so I recommend adding this data in the text. The influence of the inflammatory state on nutritional status and body composition and how the nutritional intervention can mitigate the inflammation should be considered.
Author Response
Dear Reviewer,
thank you for your remarks. We appreciate the suggested proposals, which we incorporated in the text.
Comments 1: In addition to malnutrition, reduced muscle mass and sarcopenia are also considered modifiable risk factors that can be improved with nutritional therapy. I recommend adding these elements in the introduction section.
Response 1: Lines 25-27: Additionally, reduced muscle mass, sarcopenia, and sarcopenic obesity are considered to be modifiable risk factors as well (2).
Comments 2: Line 62, page 2: add reduced muscle mass not only sarcopenia
Response 2: Lines 63-65: Sarcopenia, reduced muscle mass, and malnutrition have been shown to negatively impact the postoperative course regarding the occurrence of complications and length of hospital stay (3,9,10).
Comments 3: In "Unimodal nutritional prehabilitation" section specify the type of surgery performed in the studies cited.
Response 3: Line 86/87: Meta-analysis of patients undergoing major abdominal surgery
Line 96: (…) with patients undergoing colorectal surgery (…)
Line 102: patients with colorectal cancer
Line 107: (…) including patients undergoing any surgery (…)
Comments 4: The bibliography and consequently the text must be updated with recent systematic reviews such as: Mareschal et al. Cancer (2023), Jain et al. World J Surg (2023), Pavel Skorepa et al. Clinical Nutrition (2024)
Response 4: Thank you for that suggestion. We added the named literature in line 116-120: As demonstrated in recent systematic reviews and meta-analyses multimodal prehabilitation will primarily improve functional capacity and physical performance before surgery (Mareschal et al, 2023, Jain et al, 2023) . A significant decrease of complication rate may be awaited with special regard to frail and high risk patients (Skorepa et al, 2024).
Comments 5: In most published studies regarding surgical prehabilitation, the PG-SGA is used as a method of malnutrition assessment. Add it in the "Assessment of nutritional status and indication for prehabilitation" section. Body composition assessment and measurement methods must be clarified better in the "Assessment of nutritional status and indication for prehabilitation" section.
Response 5: Thank you, that is totally true. We completed this in lines 160-161: Furthermore, the Subjective Global Assessment (SGA) or the Patient-generated SGA (PG-SGA) is used for malnutrition screening and assessment in clinical studies.
Lines 177-184: In an axial CT scan at the lumbar level 3 during the portal vein phase, muscle mass and muscle quality can be measured using various software programs based on tissue-specific attenuation values for skeletal muscle (-29 to 150 Hounsfield Units) (Gomez-Perez et al.). The skeletal muscle area in this slice includes seven muscle groups: the psoas, erector spinae, quadratus lumborum, transversus abdominis, external obliques, internal obliques, and rectus abdominis. The Skeletal Muscle Index (SMI), which is used for diagnosing sarcopenia, can be easily calculated by dividing the skeletal muscle area by height squared.
Comments 6: I recommend adding ONS containing whey proteins in table 1.
Response 6: Thank you for your suggestion, whey protein has been added to table 1.
Comments 7: I recommend separating table 1 into two tables: a table relating to the composition of ONS and a table relating to Textures and flavours (the powdered ONS should also be added).
Response 7: Thank you for the constructive criticism – we have separated the tables and added powder.
Comments 8: Vitamin supplements should also be considered in case of deficiencies (vitamin D, B12, folate...) as they are part of the nutritional intervention.
Response 8: Lines 248-250: Additionally, in case of deficiency, supplementation of vitamins and micronutrients can be considered. Most importantly to mention are vitamin D, folat, vitamin B12. Some studies provided multivitamin supplements during prehabilitation (12).
Comments 9: Most of the published studies included cancer patients so I recommend adding this data in the text. The influence of the inflammatory state on nutritional status and body composition and how the nutritional intervention can mitigate the inflammation should be considered.
Response 9: Lines 37-41: Additionally, in cancer patients, the carcinoma significantly affects immunomodulating processes, negatively impacting nutritional status and body composition. Therefore, most studies on prehabilitation programs focus on cancer patients undergoing surgery to mitigate inflammation processes.
Reviewer 3 Report
Comments and Suggestions for Authors
This is an excellent review article which is written by experts and provides guidelines for nutritional prehabilitation in patients (particularly those with cancer) undergoing abdominal surgery.
Some minor comments:
The methodology section briefly mentions a literature search conducted in September 2023 using PubMed with terms like "Prehabilitation," "Nutrition," "ONS," and "Compliance," but it lacks detailed information on the search strategy. Please provide a more comprehensive description of the methodology, including the databases searched, specific inclusion and exclusion criteria, the number of articles retrieved, and how they were selected for inclusion in the review. This will enhance the reproducibility and transparency of your review.
The manuscript presents results from various studies and meta-analyses, but the reporting is inconsistent. For instance, while some sections provide specific data and confidence intervals, others offer general conclusions without detailed statistics. Please ensure consistency in reporting study outcomes by including specific statistics, effect sizes, confidence intervals, and p-values where available. This will help readers better understand the strength and significance of the findings.
Author Response
Dear Reviewer,
thank you for your remarks. We appreciate the suggested proposals, which we incorporated in the text.
Comments 1: The methodology section briefly mentions a literature search conducted in September 2023 using PubMed with terms like "Prehabilitation," "Nutrition," "ONS," and "Compliance," but it lacks detailed information on the search strategy. Please provide a more comprehensive description of the methodology, including the databases searched, specific inclusion and exclusion criteria, the number of articles retrieved, and how they were selected for inclusion in the review. This will enhance the reproducibility and transparency of your review.
Response 1: We did not conduct a systematic literature search as this is a narrative review. However, we focused on randomized controlled trials (RCTs), meta-analyses, and systematic reviews. We have amended the following paragraph to make this clearer:
Lines 49-55: Methodology: For this narrative review, , a non-systematic literature search was conducted on PubMed September 2023 using the terms "Prehabilitation" and "Nutrition," as well as "ONS" (Oral Nutritional Supplements) and "Compliance." We primarily included randomized controlled trials (RCT), meta-analyses, and systematic reviews. Emphasizing psychological aspects, several recommendations for clinical practice were formulated.
Comments 2: The manuscript presents results from various studies and meta-analyses, but the reporting is inconsistent. For instance, while some sections provide specific data and confidence intervals, others offer general conclusions without detailed statistics. Please ensure consistency in reporting study outcomes by including specific statistics, effect sizes, confidence intervals, and p-values where available. This will help readers better understand the strength and significance of the findings.
Response 2: Thank you for this remark. This is totally correct and we added some more specific statistics in some studies: Line 73, Line 98-99, Lines 114-116, Lines 134-135, Lines 212-214, Line 223.
Reviewer 4 Report
Comments and Suggestions for Authors
1. The main question addressed by the research
The primary question addressed by the research is whether dietary prehabilitation is helpful and practicable in individuals having major abdominal surgery. The study's goal is to synthesise existing research, identify barriers and facilitators, and make practical suggestions for clinical practice.
2. Originality and relevance in the field
The topic of nutrition prehabilitation is both novel and timely in the field of surgical treatment, particularly for patients having significant abdominal procedures. It fills a unique need in the area by emphasising the diversity of nutritional treatments and the scarcity of unambiguous, synthesizable data for specific dietary recommendations.
3. Specific improvements in methodology
The authors might include a more extensive discussion of the methodology they used to identify and analyse literature by creating a Materials & Methods section. This involves establishing inclusion and exclusion criteria, database searches, and data extraction and analysis methods.
4. Consistency of conclusions with evidence and arguments
The conclusions drawn are typically in line with the data and argumentation offered. They clearly summarise the benefits of nutritional prehabilitation as well as the limitations of implementing it. To support their results, the authors should present more thorough evidence linking particular dietary therapies to improved surgical outcomes.
5. Appropriateness of references and comments on tables and figures
The references are current, relevant, and suitable to the study's context.
Tables: The tables are well-organized and contain detailed information about the various types of ONS and their respective applications.
Figures: Figures such as the overview of issues and facilitators to nutritional prehabilitation are useful. The concept of several patient trajectories (Figure 2) is especially valuable.
Comments on the Quality of English LanguageMinor editing of the English language is required.
Author Response
Dear Reviewer,
thank you for your remarks. We appreciate the suggested proposals, which we incorporated in the text.
Comments 1: The authors might include a more extensive discussion of the methodology they used to identify and analyse literature by creating a Materials & Methods section. This involves establishing inclusion and exclusion criteria, database searches, and data extraction and analysis methods.
Response 1: We did not conduct a systematic literature search as this is a narrative review. However, we focused on randomized controlled trials (RCTs), meta-analyses, and systematic reviews. We have amended the following paragraph to make this clearer:
Lines 49-55: Methodology: For this narrative review, , a non-systematic literature search was conducted on PubMed September 2023 using the terms "Prehabilitation" and "Nutrition," as well as "ONS" (Oral Nutritional Supplements) and "Compliance." We primarily included randomized controlled trials (RCT), meta-analyses, and systematic reviews.
Comments 2: The conclusions drawn are typically in line with the data and argumentation offered. They clearly summarise the benefits of nutritional prehabilitation as well as the limitations of implementing it. To support their results, the authors should present more thorough evidence linking particular dietary therapies to improved surgical outcomes.
Response 2: Unfortunately, specific recommendations for dietary therapies during prehabilitation cannot be made due to heterogeneous and inconsistent evidence. Since most prehabilitation programs are multimodal, drawing conclusions about the effect of nutrition therapy alone is challenging. Studies with a specific and well-explained nutrition therapy and synchronized outcome reporting are needed to enable sensible evidence synthesis, for example in meta-analysis.
Round 2
Reviewer 2 Report
Comments and Suggestions for Authors
Accept in present form